# The Influence of Different Forest Characteristics on Non-point Source Pollution: A Case Study at Chaohu Basin, China

**DOI:** 10.3390/ijerph17051790

**Published:** 2020-03-10

**Authors:** Hao Cheng, Chen Lin, Liangjie Wang, Junfeng Xiong, Lingyun Peng, Chenxi Zhu

**Affiliations:** 1Co-Innovation Center for Sustainable Forestry in Southern China, Nanjing Forestry University, Nanjing 210037, China; chenghao95114@163.com (H.C.); pawangwlj@outlook.com (L.W.); lingyunp@163.com (L.P.); 2Priority Academic Program Development of Jiangsu High Education Institutions (PAPD), Nanjing Forestry University, Nanjing 210037, China; 3Key Laboratory of Watershed Geographic Sciences, Institute of Geography and Limnology, Chinese Academy Sciences, Nanjing 210008, China; xjfpanda@163.com; 4Jiangsu Institute of Land Surveying and Planning, Nanjing 210017, China; zhuchenxi1983_1983@126.com

**Keywords:** forestland characteristics, non-point source pollution, SWAT model, forest optimization configuration

## Abstract

Forestland is a key land use/land cover (LULC) type that affects nonpoint source (NPS) pollution, and has great impacts on the spatiotemporal features of watershed NPS pollution. In this study, the forestland characteristics of the Chaohu Basin, China, were quantitatively represented using forestland types (FLTs), watershed forest coverage (WFC) and forest distance from the river (DFR). To clarify the impact of forests on NPS pollution, the relationship between forestland characteristics and watershed nutrient outputs (TN and TP) was explored on a monthly scale using SWAT (Soil and Water Assessment Tool) and the period simulation was 2008–2016. The results showed that: (1) the TN and TP showed similar output characteristics and the rainy season was the peak period of nitrogen and phosphorus output. (2) Among the forestland characteristics of forestland types, watershed forest coverage and forest distance from the river, watershed forest coverage and forest distance from the river had greater effects than forestland types on the control of watershed nutrient outputs (TN and TP). (3) In different forestland types, the watershed nutrient outputs intensity remained at the lowest level when the FLTs was mixed forest, with a TN output of 1244.73kg/km^2^ and TP output of 341.39 kg/km^2^. (4) The watershed nutrient outputs and watershed forest coverage were negatively correlated, with the highest watershed forest coverage (over 75%) reducing the TN outputs by 56.69% and the TP outputs by 53.46% compared to the lowest watershed forest coverage (below 25%), it showed that in areas with high forest land coverage, the non-point source pollution load in the watershed is smaller than in other areas. (5) forest distance from the river had an uncertain effect on the TN and TP output of the basin, the forestland itself is a source of pollution, but it also has the function of intercepting pollution movement; the forest distance from the river in the range of 500–1000 m had the lowest NPS pollution. Considering the different forest characteristics and topographical factors, an optimal allocation mode of differentiated forest land was proposed, these suggestions will provide a scheme for surface source pollution prevention and control in the basin. This research gap is the basis of real forestland optimization. We may optimize the forestland layout for NPS pollution prevention and control by clarifying the internal mechanism.

## 1. Introduction

Nonpoint source (NPS) pollution is a major environmental problem that seriously affects residents’ living conditions and economic development because it is nonspecific and difficult to identify and collect. It has become a great threat to water quality and aquatic ecosystem restoration [1]. Nitrogen and phosphorus pollution account for more than 50% of the pollution load of receiving water body such as Taihu Lake and Chaohu Lake [2,3,4], and these pollutants have a serious impact on receiving water bodies according to previous studies, causing deterioration of water quality and eutrophication of water bodies [5,6,7,8]. Therefore, controlling NPS pollution is a crucial way to alleviate water pollution [9,10,11,12]. 

Surface runoff and soil erosion play an important role in nutrient transport in soil. During the process, human activity is a key element in the loss of nitrogen and phosphorus in basins [10,13,14,15]. For example, crop fertilization significantly increases nitrogen loads in watersheds and increases the risk of total nitrogen loss [14,15]. In addition, land use/land cover (LULC) changes affect the form and intensity of NPS pollution [16,17]. The high diversity of LULC results in different degrees of soil fertility and underlying surface properties, and these factors would directly affect the amount of NPS pollution and the form of loss [13,18,19]. Previous studies have also revealed that LULC changes could influence nitrogen and phosphorus losses in basins [1,20,21,22]. The types of LULC reflect the characteristics of human activities. These characteristics determine the input source of the hydrological system and have a direct impact on non-point source pollution [22,23]. Among these LULC, agricultural land, construction land and forest land are the most sensitive. The fertilization behavior of agricultural land provided excessive nitrogen and phosphorus nutrients in the watershed, and entered the water body with surface runoff in a dissolved state during rainfall [24]. The construction land is an impervious surface, and the nutrient loss caused by soil erosion is less, the main nutrient source is from domestic wastewater in residential areas [25]. 

On the other hand, more and more studies have realized that ecological land represented by forest land, wetland, has a positive interception and reduction effect on the phosphorus load generated from cultivated land and construction land [1,6,26,27]. Take forestland as an example, vegetation cover in forest areas can help increase water infiltration and reduce the possibility of soil erosion and nutrient loss [28]. However, it should be pointed out that there is some uncertainty in the study of the effects of forest land on phosphorus loading. On one hand, some researchers believe that the existence of forest land, root system, litter retention, and infiltration of forest soil can intercept and store non-point source pollution from agricultural land [21,29]. For example, the research have showed that the increase of forest area can reduce non-point source pollution [30], Liu et al., found that returning farmland to forest could effectively control regional NPS pollution using the SWAT model in the Xiangxi River watershed [26]. Furthermore, Zhao et al., revealed that the forest buffer zone of river-banks had a significant effect on the interception of agricultural nitrogen nonpoint source pollution [6]. On the other hand, some studies have shown that forest land may also become the "source" of non-point source pollution. Changes in forest vegetation types will also lead to changes in NPS pollution load [31], even in the absence of human interference, forest land may produce a large pollution load [32]. This is also the confusion currently facing related research, which needs to be vigorously resolved.

The uncertainty of forest land is mainly because most research focuses on the relationship between forest land area and non-point source output [23,33]. Because the specific characteristics of the forest land were ignored, no research was conducted at the level of the forest land mechanism, which caused the impact of the internal characteristics of the forest land, the location relationship, etc. on non-point source pollution to be unclear. In fact, different types of forest land, forest land coverage, and forest land location all have effect on the occurrence and movement of pollution loads. Studies have shown the impact of forest land vegetation types and the geographical location of forest land on water quality of runoff [34,35], however, the quantitative and comprehensive degree of these studies needs to be improved.

In this context, the scientific issues that need to be addressed in this research are: The most important objective is that how to effectively evaluate and analyze the forestland characterizes’ influence on the output of nitrogen and phosphorus surface pollutants. In order to achieve this objective, we first need to find an effective way to evaluate the load of non-point source pollution in the river basin. Moreover, based on obtaining the NPS pollution load in the basin, we quantitatively extracted the characteristic information of the forestland in the basin and studied their influence on the nitrogen and phosphorus output in the basin.

This study attempts to use hydrological model to obtain the distribution characteristics of NPS pollution load in the basin, because the basis of the entire study is the assessment of NPS load in the basin. In recent decades, many empirical models (such as SWAT, ANSWERS, and AGNPS) and physics-based models (such as RUSLE and PLOAD) have been developed and widely used to assess the space NPS pollution load under specific environmental conditions [36,37]. As a widely used physical and hydrological model, SWAT model can use the provided spatial data to simulate a variety of hydrophysical processes in a complex watershed [12,30,38,39]. This provides a good choice for us to try to evaluate the role of woodland at the watershed scale. The Chaohu basin was selected as the study site, which is the fifth largest freshwater lake in China, and NPS pollution in Chaohu Lake has become an urgent problem [40]. In addition, there are few studies using SWAT model to evaluate NPS pollution in Chaohu Basin. Therefore the objectives of the study were: Exploring the influence of key forest characteristics on the loss of nitrogen and phosphorus in the basin using a SWAT model and trying to propose a terrain-based optimal allocation model of forestland. This study will provide data support and a scientific basis for the prevention and control of forestland layout and feature optimization.

## 2. Materials and Methods

### 2.1. Study Area

Chaohu Lake Basin is located in the middle part of Anhui Province in China and has a total area of 13,350 km^2^, within a latitude of 30°58’–32°06’N and a longitude of 116°24’–118°00’E. The region is a typical subtropical humid monsoon climate, with an annual average temperature of 16.1 °C and an annual average rainfall of 1215 mm. The location of Chaohu Lake Basin was shown in Figure 1.

Uneven rainfall distribution during the year, in which the rainy season (March to September) accounted for more than 72% of the annual rainfall. The elevation of the area ranges from 1 to 1486 m above sea level. Hangbu River is one of the main rivers entering Chaohu Lake, and its catchment area is 2010 km^2^. The Hangbu River flows through the alpine hills and plains, topographical features are prominent in the basin. At the same time, the area is also the main forestland distribution area of the Chaohu Lake Basin. Therefore, this study uses the Hangbu River watershed (HB) as the study area. The main land use in HB contains forest, urban land, grassland, waters, cultivated land and bare land. The major soil types in this area are yellow brown soil and paddy soil.

### 2.2. Nitrogen and Phosphorus Output Simulation

#### 2.2.1. The Model of SWAT

The Soil and Water Assessment Tool (SWAT) model is a distributed watershed hydrological model based on GIS. It can simulate continuous time series. The hydrological process in the basin is divided into the land phase of the hydrological cycle (i.e., the part of the runoff and the slope) and the confluence stage of the hydrological cycle (That is, the river convergence part), the hydrological cycle is mainly based on the water balance equation:(1)$$SW_{t}=SW_{0}+\sum_{i=1}^{t}\left(R_{day}-Q_{surf}-E_{a}-W_{seep}-Q_{gw}\right),$$
where $SW_{t}$ is the final soil water content (mm), $SW_{0}$ is the i-th day initial soil moisture content (mm), t represents time (d), $R_{day}$ is the i-th day precipitation (mm), $Q_{surf}$ is the i-th day surface runoff (mm), $E_{a}$ is the i-th day evapotranspiration (mm), $W_{seep}$ is the i-th day the amount of water entering the vadose zone from the soil profile (mm), $Q_{gw}$ is the i-th day return to the flow of water (mm).

#### 2.2.2. The SWAT Datasets

The SWAT database input was done in ArcMap 10.3 (ERSI, Redlands, CA, USA) extended ArcSWAT 2012 (USDA, Texans, TX, USA). The basic data needed to build the model includes topography, land use, soil properties and weather data. Detailed description and source of the data are shown in Table 1.

#### 2.2.3. SWAT Model Setup

The running process of SWAT model mainly includes watershed delineator, Hydrologic Research Unit (HRU) analysis, data input, SWAT simulation. The spatial data (Digital elevation model (DEM), land use map, soil map) were projected into the same coordinate system (WGS1984) for the SWAT model setup.

Watershed Delineator is the basic step in the model operation, and the DEM data was used for watershed generation and sub-basin division. During this process, the HB was divided into 106 sub-watersheds, based on the user-defined threshold area of 1500 ha, the user-defined threshold area was the minimum catchment area of river.

Based on the sub-watershed, the SWAT model divides different areas with the same combination into the same type of HRU according to land use types, soil types and slopes, and assumes that the same type of HRU has the same hydrological behavior in the sub-watershed. In this step, land use data, soil data and slope data were prepared and entered into the model. The major land use types contain forest, urban land, grassland, waters, cultivated land and bare land, the main type of the soil contains: Cumulic Anthrosols (49%), Dystric Cambisols (21%), Eutric Planosols (10%), Haplic Luvisols (7.8%) and Cumulic Anthrosols (7.4%), the slope was reclassified into five levels, 0 ~ 7%, 7 ~ 18%, 18 ~ 31%, 31 ~ 46% and > 46%. Using the above data, the HB was further divided into 730 HRUs, HRU is the smallest unit of calculation in SWAT. Each HRU is a unique combination of land use, soil type, and terrain slope, representing homogeneous hydrological regions defined with unique land use, soil and slope.

After completing the above two steps, we need to build the model database. In this step, the specific growth habits of land-covered crops, soil nutrient attributes, etc., and climate data such as precipitation and air temperature are all consolidated and input into the model for calculation.

Parameter sensitivity analysis is an important step in SWAT simulation. The SWAT model based on the physical model mechanism requires more input parameters, and the initial parameters of the model have greater uncertainty due to parameter space differences and errors in the acquisition process [13]. Therefore, the parameter sensitivity analysis of the parameters is performed to reduce the number of model calibration parameters and improve the efficiency of model running time. In this study, Latin Hypercube One factor At a Time (LH-OAT) global sensitivity analysis parameters for the analysis sensitivity model [41]. LH-OAT assumes that each parameter will be divided into N spaces and then select the sample space of a random sample (the probability of plotting in each space is 1/N). After randomly combining all the parameters and running the model for N times, the model results were analyzed using multi-parameter linear regression or correlation analysis methods [40]:(2)$$S=\frac{\Delta R}{\Delta P_{i}}\times \frac{P_{i}}{R},$$
where S is the sensitivity index; *R* is the model outputs; $\Delta P_{i}$ is the change of model impact factor; ΔR is the change of model outputs. Table 2 shows the sensitive parameters and characteristics of this study.

Sensitivity parameters and their specific conditions were described in Table 2.

Run the model only after all the above preparations were completed. The SWAT model was carried out at the HRU level in monthly time-scale for the period 2008–2016, and a warm-up period 2007. The outcomes were aggregated to give output at the sub-watershed scale [1].

#### 2.2.4. Calibration and Validation of SWAT

The coefficient of determination (R^2^) and the Nash coefficient (Ens) between the model simulation value and the measured value were used to verify the model simulation accuracy and evaluate the reliability of the model. R^2^ represents the consistency between the simulated value and the measured value. The higher the coefficient, the better the consistency. The value of Ens is negative infinity to 1, and the closer the Ens value is to 1, the better the quality of the model and the higher the credibility of the model:(3)$$R^{2}=\frac{\sum_{i=1}^{n}(O_{I}-\bar{O})\left(S_{I}-\bar{S}\right)}{{\left[\sum_{i=1}^{n}{(O_{I}-\bar{O})}^{2}\right]}^{0.5}{\left[\sum_{i=1}^{n}{\left(S_{I}-\bar{S}\right)}^{2}\right]}^{0.5}}\;,$$
where $O_{I}$ is the measured value; $\bar{O}$ is the average of measured values; $S_{I}$ is the simulated value; $\bar{S}$ is the average of the simulated value; n is the number of the measured values or the simulated values:(4)$$\mathrm{Ens}=1-\frac{\sum_{i=1}^{n}{\left(Y_{\mathrm{obs},i}-Y_{\mathrm{sim},i}\right)}^{2}}{\sum_{i=1}^{n}{\left(Y_{\mathrm{obs},i}-Y_{\mathrm{obs},\mathrm{av}}\right)}^{2}}\;,$$ where $Y_{\mathrm{obs},i}$ is the measured value; $Y_{\mathrm{sim},i}$ is the simulated value; $Y_{\mathrm{obs},\mathrm{av}}$ is the measured average value; n is the number of observations.

The Sequential Uncertainty Fitting version 2 (Sufi2) algorithm was used to determine the results of the model. The measured flow data of the Taoxi hydrological site from 2008 to 2011 was selected as the data. The model was determined and verified based on the current land use data. Since the acquired hydrological site data was 2008–2011, the time period for the type was 2008–2010 and the verification period is 2011.

### 2.3. Characteristics of The Forest

The HB belongs to a transitional zone from hills to plains. The northern part of the basin is dominated by plains, while the southern part is covered with alpine hills. The special topography makes the forest stand characteristics diversified. The main forestland types are mixed forest, evergreen forest and deciduous forest. The forest coverage in each sub-basin is quite different. Some sub-watersheds have a forest coverage rate of more than 90%, while some sub-watersheds do not have forests. As one of the main lake systems in Chaohu Lake, and the river system in HB is relatively developed. The forestland in the upper reaches is generally located around the river, while the forestland in the lower plain is a certain distance from the river. Based on the abovementioned background, this paper used the typical forest stand characteristics of forest type, forest cover rate and the distance between forestland and river as the research variables to explore the response relationship between different stand characteristics and the nitrogen and phosphorus output of the basin.

Because this study only addressed the impact of changes in forestland characteristics on nitrogen and phosphorus output in the basin, it was necessary to ensure that other land use types did not change, which would exclude changes in the nitrogen and phosphorus output caused by changes in other land types. Under this premise, the forestland types (FLTs) were determined with SWAT’s scenario simulation function; the two characteristics of forest coverage and river distance were selected to analyze the sub-basins with corresponding characteristics. In the study of WFC and DFR, the type of forestland was unified as FRST.

To accurately simulate the effect of FLTs on the total nitrogen and total phosphorus output, the forestland types were set as mixed forest (FRST), evergreen forest (FRSE), deciduous forest (FRSD), and other land uses. Table 3 showed the key parameter settings for different forestland types (FRST, FRSE, FRSD). The types remained the same to ensure that other parameters were identical except for the forest type.

The forest coverage rate refers to the percentage of forest area in a sub-basin to the total area of the sub-basin. The forest coverage was calculated in ArcSWAT. In this study, there were 81 sub-basins with forestland (Figure 2). To reflect the characteristics of forest coverage and nitrogen and phosphorus output more intuitively, the forest coverage rate was divided into four grades according to the average classification method.

The river network distribution in the Chaohu Lake Basin was extracted from a 90 m resolution DEM in the ArcMap 10.3 (ERSI, CA, USA) hydrological analysis template, and the extracted river network raster was converted into a vector file using the “raster to vector” function. A satellite image was used to correct the accurate river network distribution in the Chaohu Lake Basin. Then buffer zones for the river network were set up with the distances of < 500 m, 500 m~1000 m, >1000 m. According to the intersection of the vertical river in the direction of the forest and the buffer zone, the distance between the forest and river was determined. The distance between the forest and river was less than 500 m, 500 m~1000 m and greater than 1000 m.

## 3. Results

### 3.1. The Output of TN and TP

The SWAT model used sub-watersheds as a unit to output the TN and TP content of each sub-watershed. The results showed that the runoff simulation results in the calibration period were in good agreement with the measured data values (Figure 3). The R^2^ rate were greater than 0.7 and the Ens were greater than 0.5. This result indicated that the SWAT model could effectively simulate the hydrological characteristics of the HB. The verification of water quality data also achieved good performance, validated mainly for TP concentrations, and R^2^ was 0.62. Therefore, based on the land use data to establish a distributed hydrological model, the nitrogen and phosphorus output of different forestland in the HB can be simulated.

Table 4 shows the statistics of the TN and TP output of each sub-basin. In each sub-basin, the output of TN was between 14.78 and 3095.83 kg/km^2^, and the average output intensity was 1405.08 kg/km^2^; the output of TP was between 0.029 and 844.48 kg/km^2^, and the average output intensity was 394.08 kg/km^2^.

In terms of the spatial distribution, the TN and TP output in the basin was mainly from the northern region. Figure 4 shows that the nitrogen and phosphorus output intensity in the southern mountainous area was smaller than that in the northern plain area.

### 3.2. Impact of FLTs on TN and TP Output

Table 5 shows the watershed TN and TP outputs when the FLTs changed in the basin. As shown in the table, when the forestland in the basin was FRST, the annual total output of total nitrogen was at least 1244.73 kg/km^2^. In addition, when the forestland in the basin changed to FRSD, the annual total nitrogen output increased by 21.63% to 1513.98 kg/km^2^. The output of total phosphorus was the same as that of total nitrogen. The annual output of FRSD was 1.24 times that of FRST and 1.04 times that of FRSE.

It is worth noting that although there was a large difference in the total nitrogen output of the three types of forestland in terms of the annual output, the monthly total nitrogen output of the three types of forestland during most of the rainy and dry seasons were relatively close (Figure 5). Moreover, all forestland types had peak output in July (FRST 455.26 kg/km^2^, FRSE 485.88 kg/km^2^, FRSD 488.87 kg/km^2^). In the dry season, the output of the three was very small and close. The total phosphorus output had the same trend.

### 3.3. Impact of WFC on TN and TP Output

The nitrogen and phosphorus output intensity of the watershed and the forestland coverage of the watershed had a significant negative correlation (Figure 6).

When the WFC is 50% ~ 75%, the annual output intensity of total nitrogen in the forestland was 1251.43 kg/km^2^. When the WFC was between 75 ~ 100%, the total annual output of total nitrogen was only 791.26 kg/km^2^, and the output of TN and TP show the same distribution characteristics (Table 6).

During the year, the total nitrogen loss showed three peaks, April, July and September, mainly in the rainy season of the study area. In April, the total nitrogen output intensity of the sub-basin with 50 ~ 75%WFC was the largest, 0 ~ 25% of the sub-basin total nitrogen output is the lowest. In July and September, the total nitrogen output of the sub-basin with a WFC of 75 ~ 100% was smallest. The characteristics of total phosphorus loss were similar to those of total nitrogen. It is worth noting that in the peak period of total phosphorus loss, the outputs of the sub-basins with 0 ~ 25% and 50 ~ 75% WFC were the largest and close, while the total phosphorus output of the sub-basins with 25~50% and 75~100% coverage were also relatively close, and the value was small.

### 3.4. Impact of DFR on TN and TP Output

The geographical position of forestland mainly refers to the distance between forestland and river, which has the same influence on the output of TN and TP in the basin. When the position of forestland was 500 ~ 1000 m from the river, the content of the nutrient output was lowest; the total nitrogen output was 52.93 Kg/km^2^, and the total phosphorus output was 14.71 kg/km^2^. When the distance between forest and river was greater than 1000 m, the total nitrogen and total phosphorus outputs were much greater than those in the other watersheds. The total annual nitrogen output of these basins was 118.19 kg/km^2^ and the total phosphorus outputs was 31.33 kg/km^2^ (Figure 7).

The outputs of total nitrogen and total phosphorus in the watershed when the distances between the river and forest were 0 ~ 500 m and 500 ~ 1000 m were close. The total nitrogen and total phosphorus output in the watershed when the distance between the forestland and river was 0 ~ 500 m was slightly larger than that when the distance was 500 ~ 1000 m in the watershed. The maximum output of total nitrogen and total phosphorus in these two types of watersheds was in September, and the total nitrogen and total phosphorus output was lowest in November. In the basin where the forest was 1000 m from the river, the total nitrogen total phosphorus output reached its maximum in July, and the output was the lowest in November (Figure 8).

## 4. Discussion

### 4.1. The Response of Forest Characteristics to TN and TP

Previous studies have focused relatively more on explaining the composition of watershed land cover to explain changes in nutrients [23], but there are not many studies on the impact mechanisms of specific land cover internal characteristics. However, the absorption and interception of nutrients by forests can effectively reduce the output of watershed pollutants [42]. This study deeply explored the characteristics of forestland to explore the mechanism of forestland characteristics on the nitrogen and phosphorus output intensity of the basin.

The results of this study indicated that the output intensity of TN and TP in the basin were consistent, but the output intensity of total phosphorus was lower than that of total nitrogen. The rainy season output intensity of nitrogen and phosphorus pollutants was much higher than that in the dry season, mainly due to the severe mineralization process of litter in the rainy season. Therefore, in the rainy season, the polymorphism of nitrogen and phosphorus in the litter is more easily washed into the river by rainfall [43,44], and the rainy season also has more abundant precipitation than the dry season, resulting in more surface runoff [45]. Gu, et al. [46] and Wilson [47] also proved that the phosphorus dynamics in soil was strongly influenced by climatic factors.

• *How the hydrological characteristics of different FLTs and the degree of litter mineralization of affect the nutrient output intensity*.

Compared with FRSE and FRSD (Figure 5), FRST had smaller nitrogen and phosphorus output intensity, indicating that mixed forests have more effects especially the prominent absorption and interception on nitrogen and phosphorus nutrients. This result is consistent with the conclusion of Sprenger et al., that the FRST obtained in the study performed better at preventing nutrient loss [48]. As an important carrier of nutrient salt outputs such as nitrogen and phosphorus, water directly affects the migration and transfer process of nutrients such as nitrogen and phosphorus [21,23]. Liu et al., found a significant positive correlation between runoff and nitrogen and phosphorus loss, and the degree of feedback from different forest types to forest hydrology also led to differences in nitrogen and phosphorus output intensity [49]. Mixed forests have the characteristics of multiple forests. The canopy can effectively intercept precipitation and reduce the kinetic energy of raindrops [34]. The lower litter layers increase the surface coverage [50], buffer the splashing kinetic energy of raindrops that penetrate the forest, increase soil water infiltration and reduce surface runoff. The amount of nitrogen and phosphorus transported by runoff is limited, which directly reduces the output intensity of nitrogen and phosphorus. Compared with mixed forests, evergreen and deciduous forests represent single forest types. Although the evergreen forest had obvious advantages in intercepting the canopy, the surface cover was lower. When rainfall reach the canopy to and eventually the ground, certain degree of surface runoff results [51]. As shown in Figure 4, the nitrogen and phosphorus output of the FRSD was similar to the FRSE pattern during the rainy season, but there were significant differences between March and April. Drought mainly occurs throughout winter, and when rain occurs in March, the litter layer of the deciduous forest can intercept and block the occurrence of surface runoff, however, due to the polymorphism of nitrogen and phosphorus compounds in the litter, nitrogen and phosphorus mineralization occurs [43]. The mineralized nitrogen and phosphorus are easily washed away by rainwater and directly enter the river in runoff, increasing the nitrogen and phosphorus output intensity of the basin.

• *Forests can control the production of NSP pollutants and simultaneously intercept the migration of NSP pollutants*.

An increase in forest area contributes to water, infiltration, reducing the possibility of soil nutrient loss. The results of this study also prove this point. Figure 6 shows that when the forest area was greater than 75%, the nitrogen and phosphorus output intensity was the lowest. The hydrological process is an important mechanism to explain the changes in NSP pollutants in the basin [14]. The increase in forests reduces the total flow at the basin scale. The large number of NPS in the soil lacks transport carriers [39], reducing the risk of pollutant output. Cecílio et al., pointed out that in areas with high forest cover, the water flow is relatively more stable and sustainable [35], which also explains why the annual output intensity of nitrogen and phosphorus was stable and in the sub-basin with 75% forest coverage in this study. However, in this study, when the forest coverage rate was between 50–75%, the nitrogen and phosphorus output intensity was relatively higher intensity. Figure 1 shows that this part of the sub-basin was mainly located in the transition zone from mountain to plain, and the area not covered by forest was reclaimed as sloping land. Studies have proven that sloping farmland is an important source of nitrogen and phosphorus output [26,42], which directly increases the nitrogen and phosphorus output intensity of these sub-basins. This finding also proves the research results of Cecilio et al. [35]. The impact of forest cover on the nitrogen and phosphorus output of the basin was mainly shown in the large watersheds, and in the smaller watershed, the impact of forest cover was uncertain [52].

• *Forest buffers could trap, infiltrate, adsorb, and convert contaminants*.

NPS pollution is mainly divided into process and migration factors [53]. The extensive and deep roots and highly permeable soils in forest areas provide nutrient flow protection, forming a sink. Contaminants are trapped, infiltrated, adsorbed and converted as they flow through forest buffers via surface runoff [53]. However, few scholars have quantitatively studied the influence of the distance between forest and river on the output intensity of nitrogen and phosphorus. This study explored the influence of the position of forestland and river on the nitrogen and phosphorus output in a single independent catchment unit. The forestland around the river had a significant reduction effect on the amount of nitrogen and phosphorus in the river. The nitrogen and phosphorus output intensity of the forestland within 1000 m of the river reduced the TN output by 55.22% and the TP output by 53.48% compared with the forestland that was 1000 m away. However, there is little doubt here why the nitrogen and phosphorus output intensity when the forest distance was less than 500 m was not lowest. There are three main reasons. First, in the hot and humid environment during the rainy season, the mineralization process of the litter in the forest is intense. When rainfall occurs during the rainy season (March–September), polymorphic nitrogen and phosphorus in the litter are more easily washed by rain and enter the river [43,54], as shown in Figure 8, and the difference in the nitrogen and phosphorus output intensity is large. Moreover, the second reason is related to the location. The runoff on steep terrain is relatively more serious in terms of soil erosion, resulting in more soil entering the river. Vilmin et al. [55] pointed that slow runoff increased the accumulation of phosphorus, and polymorphic nitrogen and phosphorus are lost and eventually enter the river with water and sediment. Third, the balance of forest infiltration and evaporation has a direct effect on the nitrogen and phosphorus output transport capacity of the basin [30].

### 4.2. Optimized Allocation of Forestland

At present, most studies have adjusted the land use layout to obtain the best management model for NPS pollution prevention and control, but there has been less consideration regarding the internal characteristics of forestland. The existing research can only determine the number of areas of forestland in the best management mode, and the location relationship with other land types; how the characteristics of the forest interior should be configured is not clear. This study tentatively proposes an optimal allocation model based on three types of FLTs, WFC and DFR. At the same time, considering the differences in the ecological and economic functions between mountainous and plain areas, a differentiated allocation model based on the three elements is proposed based on the topography.

The results of the study showed that FRST had a relatively lower nitrogen and phosphorus output intensity while the other characteristics remained unchanged (Table 5). Therefore, this configuration used FRST as the main FLTs. Based on clarifying the type of forest stand, we further explored the allocations of WFC and DFR in different terrain conditions. HB was divided into mountainous areas and plain areas. The sub-basin in each area was divided into low-intensity, medium-intensity and high-intensity output according to the natural break method. The optimal allocation of forestland under different terrains was investigated by studying the allocation of forestland in low-intensity sub-basins. Table 7 shows the status of forestland allocation in the low-intensity sub-basins with the nitrogen and phosphorus outputs in the mountainous and plain areas.

Mountain areas are recommended for high-coverage forests; they should be densely planted and in close proximity to river areas. In the sub-basins with the lowest nitrogen and phosphorus output intensity (Table 7), most of the sub-basin forest coverage exceeded 75%, and the distance from the river was also within 500 m. Bonnesoeur et al., found that the soil erosion intensity of large forests on steep slopes was relatively lower [27]. Therefore, when forestland was used in the mountainous area, it tended to maintain the water and soil conservation water source function, which requires high coverage, especially in areas with steep slope, to reduce the nitrogen and phosphorus output caused by soil erosion via runoff [56].

It is not necessary to plant high-coverage forestland in the plain area, but afforestation should be carried out near the water outlet of the cultivated land. It is also recommended that a forest buffer zone be placed at different distances from the river. Cultivated land in the plain area of the study area was the main type of land cover, and the distribution of forestland was lower and scattered. Therefore, for this reason, the forest coverage of the sub-basin with the lowest nitrogen and phosphorus loss in the plain area was less than 25%; Table 7 shows that more than 50% of the sub-basin forestland with a low-intensity nitrogen and phosphorus output was within 500 m of the river. Although the coverage of forestland was low, the distance between the forestland and river was relatively close, and the forestland was similar to the existence of the buffer zone. Zhang et al., found that the existence of a river vegetation buffer could reduce the TN (by 13.94%) and TP (by 9.86%) entering a river, reducing the risk of nitrogen and phosphorus output in the basin [57]. This result suggests that the allocation of forestland in the plain should focus on the buffer function to intercept and absorb the nitrogen and phosphorus flowing through the forest [27]. The results in Table 6 also show that increasing the forest coverage had a positive effect on reducing nitrogen and phosphorus. Appropriately increasing the forest coverage to reduce the nitrogen and phosphorus output intensity also needs to be considered. However, due to the developed agriculture in the plain area, the soil fertility is also good. Large-scale afforestation is obviously inappropriate, and therefore, large-scale afforestation along the river is a good choice. Riverside shelterbelts should be constructed, riverside soil should be consolidated, and surface runoff should be intercepted.

Therefore, comprehensively considering the layout pattern of mountains and plains, in the HB, our proposed model in the mountainous area is FRST + WFC (>75%) + DFR (<500 m). the plain area is a little different: FRST + WFC (>25%) + DFR (<500 m). The results of the optimized pattern of forestland layout were shown in Figure 9. We optimized the configuration mode from mountain and plain terrain.

### 4.3. Research Issues and Prospects

Our results showed that the specific characteristics of forestland in the watershed have a certain effect on the regulation of NPS pollution in the watershed. The research of some scholars in other regions also proves our point of view. Such as Cecílio et al. [35] proved that the location of forestland could influence the streamflow. Gu et al. [46] pointed out that lithological and bioclimatic had impact on soil phosphatase activities in California temperate forests. These studies all revealed that the characteristics of land use have a significant impact on nitrogen and phosphorus output.

When studying the impact of coverage and forest location, disturbances in terrain conditions were found. Topographical characteristics influence nutrient transport pathways [58], and relatively higher slope variability leads to higher flow and erosion intensity, increasing the amount of nutrient outputs in the basin [59]. The feedback of nitrogen and phosphorus output in forestland under different topographic conditions deserves further study.

Under the same topographic conditions, the nitrogen and phosphorus output of the watershed with the same forestland characteristics, forest coverage and geographical location still had large differences. Studies have shown that the landscape pattern of watershed cover has an important impact on the hydrological cycle and nutrient pollution process of a basin [21,60,61]. Lee et al. (2009) and Shi et al., found that large unbroken forests showed greater water purification potential and improved the water quality [8,22]. In this study, affected by topographic conditions, economic development and agricultural farming, the distribution of forestland in the transition zone of hills and plains was not concentrated, and the size of forest plaques also differed. These factors may have resulted in the output intensity of nitrogen and phosphorus in the basin having the same characteristics as forestland. Different dominant factors and the actual impact capacity of the forest landscape pattern will be the focus of the next phase. Of course, the focus of this study is the output characteristics of NPS pollution load in each season of the year. To more accurately show the influence of forestland characteristics on the nitrogen and phosphorus output of the basin, long-term research can be carried out in subsequent research. The next study will compare and analyze the impact characteristics of forestland features on watershed nutrients at different time scales.

## 5. Conclusions

Our research focused on the control of NPS pollution by different forestland characteristics. Considering that previous studies only focused on local watershed forestland scale, our study quantitatively and concretely studied the influence of forestland characteristics on the output of nitrogen and phosphorus surface pollutants. In addition, a differentiated forestland optimal allocation model was proposed according to the control of pollutants in forestland under different terrain conditions.

Results from simulation showed that:(1)SWAT was able to simulate the monthly nutrients outputs after the calibration with great performance in HB, and the TN total and TP showed similar output characteristics.(2)Among the three forest feature selections of FLTs, WFC and DFR, the effects of WFC and DFR on nutrient output in the basin are greater than FLTs. The FRST had lowest watershed nutrients outputs (TN and TP), the WFC had negative correction with watershed nutrients outputs, higher of the WFC, the lower of outputs, DFR had an uncertain effect on the TN and TP output of the basin.(3)Based on FLTs, WFC, DFR, the optimal allocation model of forestland is proposed. High coverage forest near the river are recommended for planting in the mountain area, and the forest zone within 500 m of the river was advised in the plain area, which will provide a scheme for basin surface source pollution prevention and control.

In general, the forest characteristic is a crucial point to be considered for the effect of watershed nutrients outputs (TN and TP). This study helps to demonstrate that, in order to obtain a more refined and deeper understanding of the control of NPS pollution, it is necessary to take into consideration the specific characteristics of forestland such as the internal structure, geographical features and natural conditions of forestland.

## Figures and Tables

**Figure 1 ijerph-17-01790-f001:**
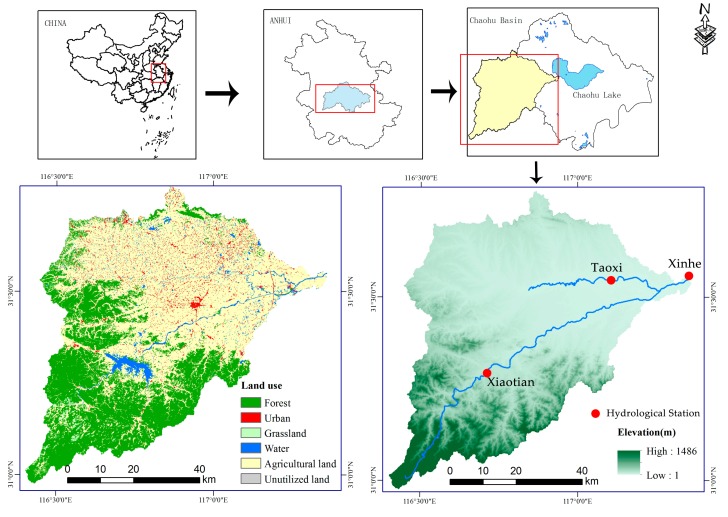
Locations of study area and samples.

**Figure 2 ijerph-17-01790-f002:**
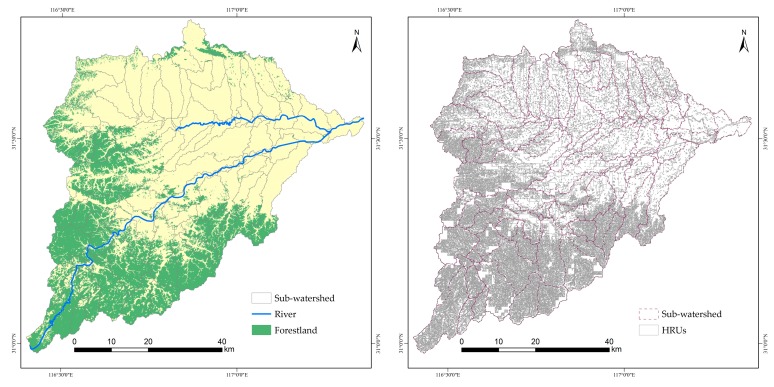
The distribution of sub-watersheds and HRUs.

**Figure 3 ijerph-17-01790-f003:**
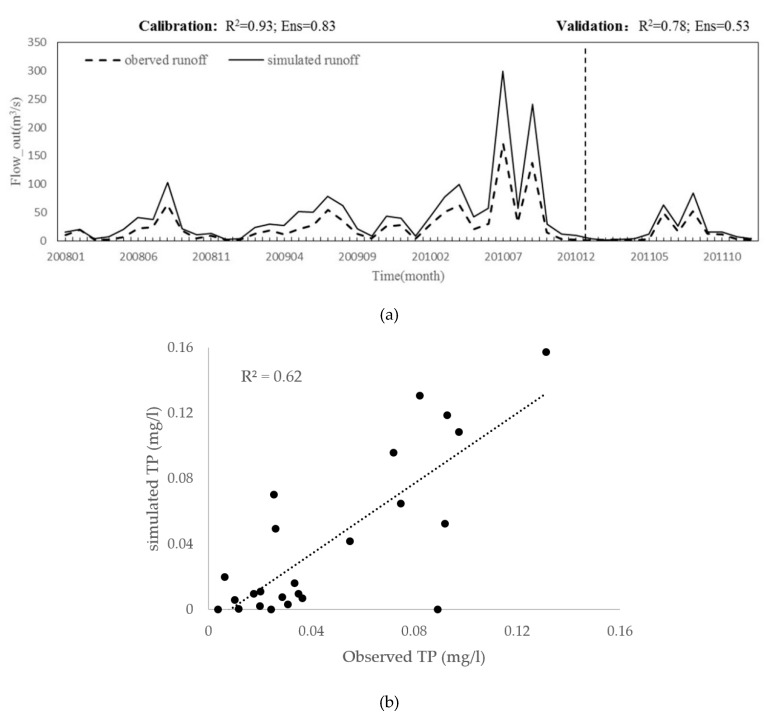
River flow correction and verification results and verification of water quality simulation (**a**): River flow correction and verification, (**b**): verification of water quality simulation).

**Figure 4 ijerph-17-01790-f004:**
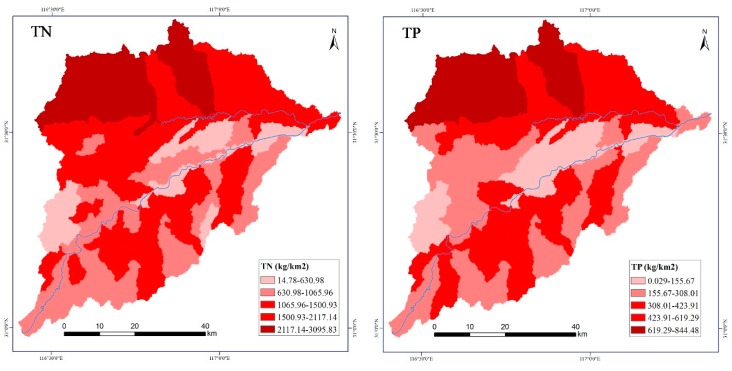
The spatial distribution characteristics of TN and TP.

**Figure 5 ijerph-17-01790-f005:**
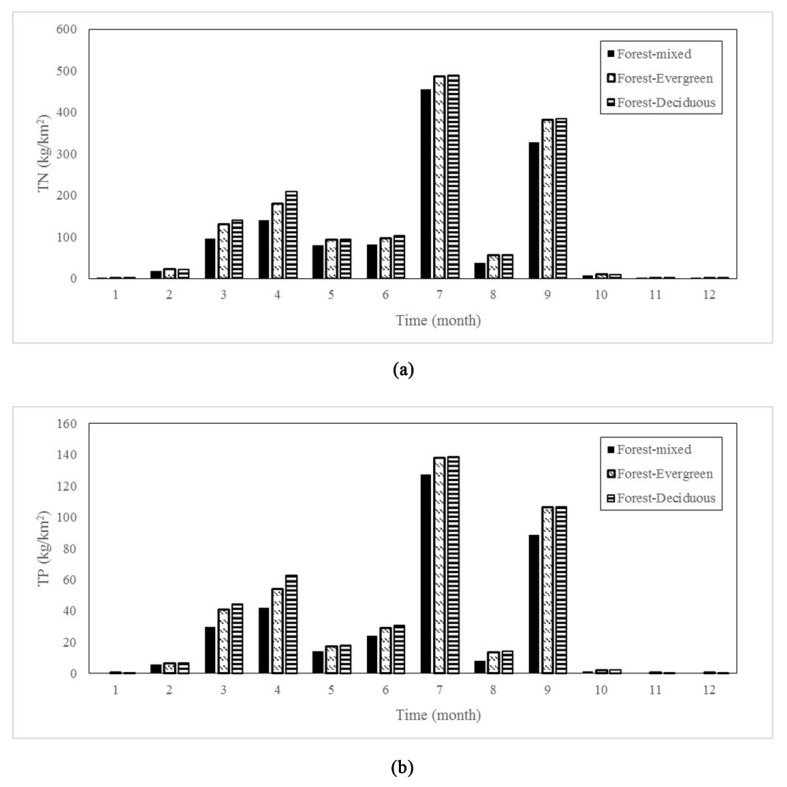
Monthly distribution characteristics of nutrients outputs with different FLTs: (**a**) was the distribution of TN; (**b**) was the distribution of TP.

**Figure 6 ijerph-17-01790-f006:**
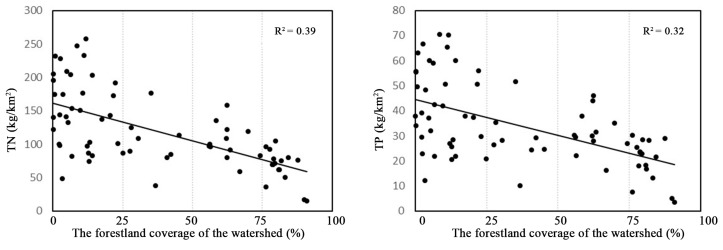
The relationship between nutrients outputs (TN and TP) and WFC.

**Figure 7 ijerph-17-01790-f007:**
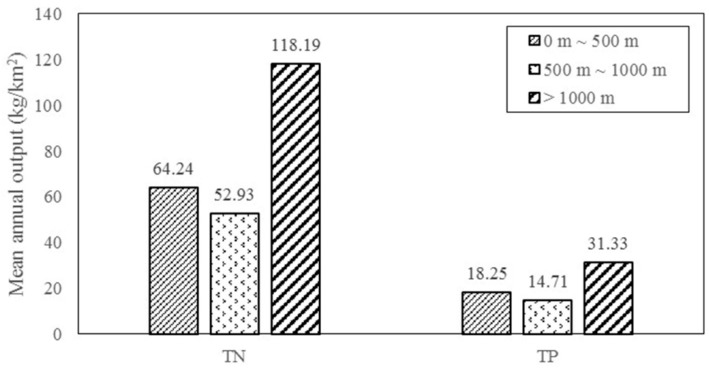
Annual output intensity of TN and TP in different DFR.

**Figure 8 ijerph-17-01790-f008:**
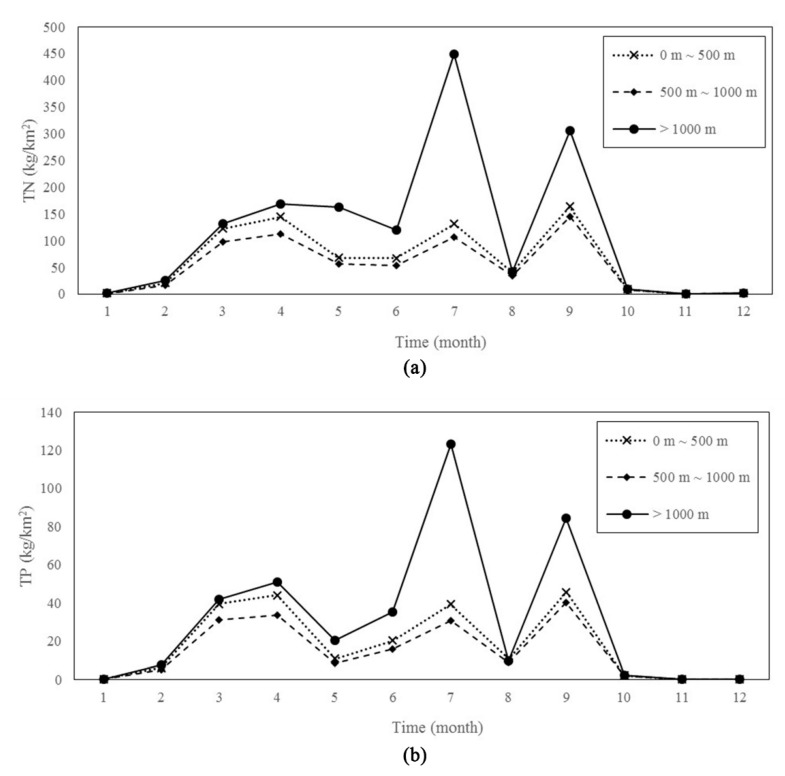
Monthly distribution characteristics of nutrients outputs with different FLTs: (**a**) was the distribution of TN; (**b**) was the distribution of TP.

**Figure 9 ijerph-17-01790-f009:**
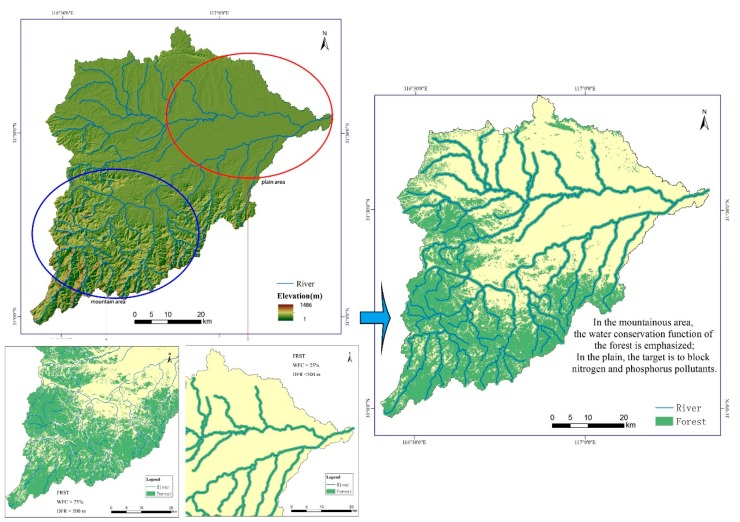
Optimized allocation of forest land oriented to NPS prevention.

**Table 1 ijerph-17-01790-t001:** Overview of data source of model running.

Data	Time	Resolution	Source
Digital elevation model		90 m	Shuttle Radar Topography Mission (SRTM). The data was obtained from United States Geological Survey (USGS).
Land use	2015	30 m	Data Center for Geography and Limnology Science, Chinese Academy of Science, CAS (http://lake.geodata.cn).
Soil properties	1987–2004	1 km	Harmonized World Soil Database (HWSD), which was built by The Food and Agriculture Organization of the United Nations (FAO) and the Vienna International Institute for Applied Systems (IIASA).
Climate data	2007–2016	1/4 degree Day by day	The China Meteorological Assimilation Driving Datasets for the SWAT model (CMADS) The data set is provided by Cold and Arid Regions Sciences Data Center at Lanzhou (http://westdc.westgis.ac.cn)
Hydrological data	2008–2011, 2012–2013	Month by month	The database including Taoxi and Xiaotian hydrological monitoring station measured stream data (2008–2011) and Xinhe monitoring station measured monthly water quality data (2012–2013).

**Table 2 ijerph-17-01790-t002:** Runoff simulation sensitivity parameter.

Runoff Parameter	Description	The Range of Values
CN2	SCS runoff curve coefficient	−0.2~0.2
ALPHA_BF	Base flow alpha coefficient	0~1
GW_DELAY	Groundwater lag time	30~450
GWQMN	Shallow depth of water	0~2
GW_REVAP	Groundwater evaporation coefficient	0~0.2
ESCO	Plant absorption compensation coefficient	0.8~1.0
CH_N2	River Manning coefficient	0~0.3
CH_K2	Effective channel conductivity	5~130
SOL_AWC	Soil available water	−0.2~0.4
SOL_K	Soil saturated hydraulic conductivity	−0.8~0.8

**Table 3 ijerph-17-01790-t003:** Key parameter settings for different forestland types.

Parameter	Description	FRSD	FRSE	FRST
FRGW1	Fraction of total potential heat units corresponding to the 1st point on the optimal leaf area development curve (dimensionless)	0.05	0.15	0.05
FRGW2	Fraction of total potential heat units corresponding to the 2nd point on the optimal leaf area development curve (dimensionless)	0.4	0.25	0.4
LAIMX1	Fraction of the maximum leaf area index corresponding to the 1st point on the optimal leaf area development curve (dimensionless)	0.05	0.7	0.05
LAIMX2	Fraction of the maximum leaf area index corresponding to the 2nd point on the optimal leaf area development curve (dimensionless)	0.95	0.99	0.95
MAT-YRS	Number of years required for tree species to reach full development (years)	10	30	50
T-BASE	Base temperature for plant growth (°C)	10	0	10
CHTMX	Maximum canopy height (m)	6	10	6
WYSE	Lower limit of receiving index	0.01	0.6	0.01
CN2A	Initial SCS runoff curve number for moisture condition II n soil hydrological unit A of HRUs	45	25	36
CN2B	Initial SCS runoff curve number for moisture condition II n soil hydrological unit B of HRUs	66	55	60
CN2C	Initial SCS runoff curve number for moisture condition II n soil hydrological unit C of HRUs	77	70	73
CN2D	Initial SCS runoff curve number for moisture condition II n soil hydrological unit D of HRUs	83	77	79

Notes: FRSD - Deciduous Forest; FRSE - Evergreen Forest; FRST - Mixed Forest.

**Table 4 ijerph-17-01790-t004:** Statistic of TN and TP output in HB.

Nutrients	Max (kg/km^2^)	Mean (kg/km^2^)	Min (kg/km^2^)
TN	3095.83	1405.08	14.78
TP	70.37	394.08	0.002

**Table 5 ijerph-17-01790-t005:** Annual output intensity of TN and TP in different FLTs.

Type of Forest	TN (kg/km^2^)	TP (kg/km^2^)
FRST	1244.73	341.39
FRSE	1458.68	407.36
FRSD	1513.98	423.78

Notes: FRSD - Deciduous Forest; FRSE - Evergreen Forest; FRST - Mixed Forest.

**Table 6 ijerph-17-01790-t006:** The output of TN and TP in the watershed with different WFC.

WFC (%)	TN (kg/km^2^)	TP (kg/km^2^)
0–25	1827.23	507.92
25–50	1224.09	345.52
50–75	1251.43	377.80
75–100	791.26	236.39

**Table 7 ijerph-17-01790-t007:** Status of forestland allocation in low-intensity sub-basins with TN and TP output in mountainous and plain areas.

Region	Nutrients	Allocation of Forest Land	Output Intensity (kg/km^2^)	Number of Sub-Basins
Mountain area	TN	FRST + WFC D + DFR A	14.47–433.04	5
		FRST + WFC A + DFR B	41.76	1
	TP	FRST + WFC D + DFR A	0.03~215.83	8
		FRST + WFC A + DFR B	7.79	1
Plain area	TN	FRST + WFC A + DFR A	241.34–1035.10	7
		FRST + WFC A + DFR B	421.58–584.32	3
		FRST + WFC A + DFR C	323.68–1213.65	6
		FRST + WFC B + DFR A	460.05	1
	TP	FRST + WFC A + DFR A	36.46–262.37	7
		FRST + WFC A + DFR B	94.62–147.28	2
		FRST + WFC A + DFR C	63.39–297.31	7
		FRST + WFC B + DFR A	121.47	1

Notes: FRST - Mixed Forest; WFC - watershed forest coverage; WFC A - the WFC of 0~25%; WFC B - the WFC of 25–50%; WFC C - the WFC of 50–75%; WFC D - the coverage of 75–100%; DFR - forest distance from the river; DFR A - the DFR within 500 m; DFR B - the DFR of 500–1000 m; DFR C - the DFR more than 1000 m.

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
