# Peer review of "The Influence of Different Forest Characteristics on Non-point Source Pollution: A Case Study at Chaohu Basin, China"

_ijerph, 2020, doi:10.3390/ijerph17051790_

Round 1

Reviewer 1 Report

There have been noticeable improvements from the current version over the previous one. The authors answered almost all questions. Therefore, I believe that this manuscript has potential to be published.

Author Response

Point 1: There have been noticeable improvements from the current version over the previous one. The authors answered almost all questions. Therefore, I believe that this manuscript has potential to be published.

Response 1: Very thanks for the reviewer’s approval.

Reviewer 2 Report

Overall, well written paper about nutrient loading in a watershed. Need to add more context and more appropriate literature on controls on nutrients.

Abstract:

Lots of acronyms, hard to keep track of

Highlighting?

L87: Not sure what this objective means

L97: appropriateness of SWAT model unclear relative to the objectives, which are themselves unclear

L456Figure 9 a result in discussion?

Should incorporate other examples and or talk about applicability of results

Wilfire? Landuse? Lithology impacts on nutrient loading? Lots of examples from Pacific Northwest of USA and also coastal California.

Wilson, Stewart G., et al. "Influence of Climate and Lithology on Soil Phosphorus." AGU Fall Meeting Abstracts. 2016.

Gu, Chunhao, Stewart G. Wilson, and Andrew J. Margenot. "Lithological and bioclimatic impacts on soil phosphatase activities in California temperate forests." Soil Biology and Biochemistry 141 (2020): 107633.

Vilmin, Lauriane, et al. "Forms and subannual variability of nitrogen and phosphorus loading to global river networks over the 20th century." Global and Planetary Change 163 (2018): 67-85.

Author Response

Response to Reviewer 2 Comments

Point 1: Overall, well written paper about nutrient loading in a watershed. Need to add more context and more appropriate literature on controls on nutrients.

Abstract:

Lots of acronyms, hard to keep track of Highlighting?

Response 1:

Very thanks for the reviewer’s approval and kindly suggestions. Because too many acronyms make it difficult to track the key points, two changes were made in the revised draft.

First, we removed a number of abbreviations from the abstract, and replaced FLTs with forestland types, replaced WFR with the watershed forest coverage, replaced DFR with forest distance from the river. L23-35 were our revision.

Second, we highlighted the innovation of our article again at the end: This research gap is the basis of real forestland optimization. We may optimize the forestland layout for NPS pollution prevention and control by clarifying the internal mechanism. This modification is on L39.

Point 2:L87: Not sure what this objective means

Response 2:

Very thanks for the reviewer’s kindly suggestions. We agree with the reviewer's point of view.

In our articles, our most important goal is to explore how to effectively evaluate and analyze the forestland characterizes’ influence on the output of nitrogen and phosphorus surface pollutants. In order to achieve this goal, and to make the goal clearer, we have refined our objective and divided the target into two points. One is how to obtain the non-point source pollution load of the river basin, and the other is based on the non-point source pollution load, quantitatively study the effects of forest land characteristics on nitrogen and phosphorus output. We hope this can made it clear to the readers. Specific modifications are in lines 94-101.

Point 3: L97: appropriateness of SWAT model unclear relative to the objectives, which are themselves unclear

Response 3:

Thanks very much to the reviewers for his comments.

We raised one point in the revised objective, how to obtain the non-point source pollution load of the river basin. The evaluation of the pollution load from the watershed was the basis of the whole study. So when we compared different hydrological models, as a widely used physical and hydrological model, SWAT model can use the provided spatial data to simulate a variety of hydrophysical processes in a complex watershed. Specific modifications are in lines 104.

Point 4: L456:Figure 9 a result in discussion?

Response 4:

Very thanks for the reviewer’s kindly suggestions. Sorry for this distress. Figure 9 is a schematic diagram of a forestland layout optimization for the mountain and plain. We have added text to make it easier to understand. ‘Therefore, comprehensively considering the layout pattern of mountains and plains, in the HB, our proposed model in the mountainous area is FRST + WFC (>75%) + DFR (<500m). the plain area is a little different: FRST + WFC (>25%) + DFR (<500m).’ Specific modifications are in lines 470-472.

Point 5: Should incorporate other examples and or talk about applicability of results

Wilfire? Landuse? Lithology impacts on nutrient loading? Lots of examples from Pacific Northwest of USA and also coastal California.

Response 5:

Very thanks for the reviewer’s kindly suggestions. Aiming at the influencing factors of phosphorus transport in soil, we added the study of Gu, et al. (2019) and Wilson, et al. (2016) to prove that the phosphorus dynamics in soil was strongly influenced by climatic factors. (L349).

In explaining the effect of runoff erosion on nitrogen and phosphorus loss, we cited the research of Vilmin et al. (2018). (L414)

At the same time, we added a discussion on the applicability of the results. Our results showed that the specific characteristics of forestland in the watershed have a certain effect on the regulation of NPS pollution in the watershed. The research of some scholars in other regions also proves our point of view. Such as Cecílio, Pimentel and Zanetti proved that the location of forestland could influence the streamflow. Gu, Wilson and Margenot pointed that lithological and bioclimatic had impact on soil phosphatase activities in California temperate forests. These studies all revealed that the characteristics of land use have a significant impact on nitrogen and phosphorus output. (L479-485)

We hope these would increase the applicability of the results.

This manuscript is a resubmission of an earlier submission. The following is a list of the peer review reports and author responses from that submission.

Round 1

Reviewer 1 Report

IJERPH 658458 Cheng et al

The influence of different forest characteristics on non-point source pollution: a case study at Chaohu Basin, China

Overall comments

This paper presents a case study in the use of the SWAT model to determine the influence of different forest cover types and distance from river on non-point source pollution (i.e. N and P). While it is clear that a lot of data and work has gone into the study, it is not at all clear what the main data sources are and how they are used to set up and test the SWAT model. There are few clear hypotheses and a general lack of clarity in terms of aims and objectives. There may well be sufficient data and results to support a publishable paper but as currently written there are too many issues to consider the paper for publication and unfortunately I have to recommend rejection. I would however encourage the authors to consider my comments below as I have tried to suggest a constructive way forward to getting the study published.

The key weakness is that there is no review or discussion of the possible sources of NPS pollution and the interactions between LULC and transport of N and P into water bodies. The authors appear to suggest in the abstract that forests retain these nutrients, but what is the purported source? Agriculture? Urban areas? There is also frequent mention of LULC change, but no mention of how LULC may be changing in the introduction or methods – I suspect the authors may in fact be referring to spatial variability rather than change through time, but there is no mention at all of time series in the Intro or Methods. My comments only go as far as the start of the results section as it would be impossible to assess the results without an understanding of the aims, hypotheses, data and methods. Some general and specific comments up to this point are provided below.

L28-29 Abstract: issues of correlation and causation here. A high proportion of forest cover may not retain nutrients but may instead reflect a low proportion of e.g. agriculture which is a source of nutrients?

In several places it is mentioned that LULC change “causes” NPS pollution but this is overly simplistic – needs a review of N and P sources and then ways in which LULC may remove N and P. This is rather different to types of LULC that may release N and P.

Citations: is it journal practice to have both direct (author, year) referencing alongside numbered sources?

For the SWAT model, little context or justification is given for its use, and the methods go straight into rather technical aspects of parameterization and sensitivity analysis without any prior explanation. Why is this a good model for the study? What is the main purpose of applying the model? Is it not sufficient to use an empirical model based on the spatial and water quality data that you have (though these are very poorly described)? The modelling may be solid but is not adequately introduced, explained or justified.

The use of SWAT to model flows is discussed – but how does this link to the LULC datasets and how is the model parameterised for change in LULC? Where do water quality data fit in?

Table 3 provides again lots of technical detail of how forest type data can be incorporated into SWAT – but with no context, explanation, justification

Specific comments

ABSTRACT

L13 has great impacts..

L15 What previous studies? Where and when? There must be hundreds

L17 concrete not a scientific term in this context

L19 What do you mean by “clarify retention and control mechanisms”?

L21 Here you mention a monthly scale but it is not at all clear how this relates to LUCL change or water quality? No mention in methods

L26 Here you mention mixed forest and outputs of N and P (minima) with no context

L31 Distance from river (DFR) “reduces” TN and TP outputs? How? Correlation or causation?

L33 first mention of mountain and plain areas, no context

INTRO

L39 How is NPS a major problem? Examples?

L43 Very vague generalisation – where/how do N and P account for more than 50% of pollution load? How is load even quantified against different pollutants?

L49 LULC change MAY cause NPS pollution depending on the LULC types and the activities but this statement is meaningless as written

L59 So is one of your aims to explore internal structure, geog features and natural conditions? These are very vague terms and would need to be clearly defined if so?

L65 Ditto – what is meant by “clarifying the internal mechanism”?

L72 their not theirs

L78 Objective not object

L80 needs to be rephrased as an objective

L89 How is a river a lake system?

L91 What do you mean “are obvious”?

Table 1 – data descriptions are way too long for a table and should be in the text or in supplementary information.

“obtained from the geospatial cloud” is not an adequate reference/source!

Land use table entry –repeated text in two columns

Hydrological data – needs explanation and location on map – metadata?

L116-67 his text means nothing to the unfamiliar reader! What is LH-OAT? References? Purpose?

Table 2 These technical runoff parameters are not explained – what are you modelling, what are the data sources?

L125 Calibration and validation – you have not even described what the model outputs are supposed to be so this section is redundant!

L146 Major issue is that there is no source or explanation of the source of N and P output data?

L153 distribution is different to what?

L155 not a sentence

L162 impact of change? Change in what? How/why is forestland changing? Or do you mean variability between subcatchments?

Table 3 What is the purpose and source of these parameters and their values? No context provided at all!

L179 81 subcatchments – needs a figure and explanation

Reviewer 2 Report

Review of the manuscript “The influence of different forest characteristics on non-point source pollution: a case study at Chaohu Basin, China” submitted to the International Journal of Environmental Research and Public Health

The paper presents an interesting discussion about the impacts from different scenarios of forestland types (mixed forest, evergreen forest, and deciduous forest), watershed forest coverage, and forest distance from the river. The simulations focused on the loss of nitrogen and phosphorus surface pollutants over the Chaohu Basin, China. The results provided are important and valuable for policy makers and managers to enhance sustainable management of the basin.

I believe that this study has the potential to be published, after the needed adjustments. A list of recommendations is summarized in the following comments:

Major Comments

Introduction

Is the current study the first hydrological modelling approach over the Chaohu Basin, China? The authors should provide more literature review mainly about the importance of the study area.

Material and Methods

Study Area

Line 88: Suggest include dry and wet season precipitation totals as the amounts of nitrogen and phosphorus are presented in both seasons in the results section.

The model of SWAT; SWAT input data

Although input data are described, it is not clear how the model was built. More details about the model setup are needed. For instance:

The threshold of the drainage area? Classes of slopes? Definition of Hydrologic Response Units?

Calibration and validation of SWAT:

Lines 139 – 141: Did the authors used a warm-up period?

Lines 142 – 148: This paragraph and Figure 2. should be moved to the results section.

Characteristics of the forest:

Lines 153 – 154: The forest coverage in each sub-basin is quite different.

Which sub-basins? This information is provided in the results section: Lines 190 – 191: HB was divided into 106 sub-basins. The authors should move this sentence to the Material and Methods section since the scenario’s described are related to the subbasin characteristics (e.g. watershed forest coverage). Which type of forest was applied for the simulations of watershed forest coverage (WFC) and forest distance from the river (DFR) scenarios? FRSD – Deciduous Forest, FRSE – Evergreen Forest, or FRST – Mixed Forest?

Results

The results, as well as the discussion about the sensitive parameters used in the calibration process, is missing! Could the authors provide the spatial distribution of TN and TP for the simulated scenarios of FLTs (FRSD – Deciduous Forest; FRSE – Evergreen Forest; FRST – Mixed Forest), WFC, and DFR.

Discussions

Lines 274-276: “Compared with FRSE and FRSD (Figure 4), FRST had smaller nitrogen and phosphorus output intensity, indicating that mixed forests have more effects especially the prominent absorption and interception on nitrogen and phosphorus nutrients.

As shown in Table 3 the values of the runoff curve number of FRST are higher than FRSE. How the absorption of FRST is more effective than FRSE? Is the interception is related to the variation of the Leaf Area Index (LAI) values between the FLTs? Could the authors provide in the manuscript the average annual cycle LAI for each forest type simulated?

Lines 409-410: “Of course, the research time period of this study was only one year”.

Why the analysis was performed for one year whether the period calibration and validation were between 2008 and 2010?

 Minor Comments

Line 21: Could you please provide the period simulation.

Line 21: SWAT (Soil and Water Assessment Tool).

Figure 1: Please improve the resolution.

Table 1: Please provide period of land use and land cover classification.

Line 128: Typo: R2 to R2

Line 138: Please provide the location of the site in Figure 1.

Table 3.  Typo: FRGW1: 015 to 0.15

Figure 3: Please improve the resolution.

Reviewer 3 Report

line 128: Error in the super index. 

Line 260: statement "but there are still insufficient studies on the impact mechanisms of specific land cover internal characteristics", is too strict. I would avoid this mainly when a systemic review is not performed yet. 

Line 344: statement "This study tentative proposes an optimal allocation model based on three types of FLTs, WFC and DFR", brings three questions: 1) which optimization method are the authors using?, 2) Where are the results of using this optimization method? and 3) how to know whether these method is valid or no?